# Epigenetic Silencing of HER2 Expression during Epithelial-Mesenchymal Transition Leads to Trastuzumab Resistance in Breast Cancer

**DOI:** 10.3390/life11090868

**Published:** 2021-08-24

**Authors:** Babak Nami, Avrin Ghanaeian, Corbin Black, Zhixiang Wang

**Affiliations:** 1Signal Transduction Research Group, Department of Medical Genetics, Faculty of Medicine and Dentistry, University of Alberta, Edmonton, AB T6G 2H7, Canada; zhixiang.wang@ualberta.ca; 2Department of Anatomy and Cell Biology, Faculty of Medicine and Health Sciences, McGill University, Montréal, QC H3A 0G7, Canada; avrin.ghanaeianmiandoab@mail.mcgill.ca (A.G.); corbin.black@mail.mcgill.ca (C.B.)

**Keywords:** epithelial-mesenchymal transition, HER2, ERBB2, breast cancer, epigenetics, trastuzumab

## Abstract

HER2 receptor tyrosine kinase (encoded by the ERBB2 gene) is overexpressed in approximately 25% of all breast cancer tumors (HER2-positive breast cancers). Resistance to HER2-targeting therapies is partially due to the loss of HER2 expression in tumor cells during treatment. However, little is known about the exact mechanism of HER2 downregulation in HER2-positive tumor cells. Here, by analyzing publicly available genomic data we investigate the hypothesis that epithelial-mesenchymal transition (EMT) abrogates HER2 expression by epigenetic silencing of the ERBB2 gene as a mechanism of acquired resistance to HER2-targeted therapies. As result, HER2 expression was found to be positively and negatively correlated with the expression of epithelial and mesenchymal phenotype marker genes, respectively. The ERBB2 chromatin of HER2-high epithelial-like breast cancer cells and HER2-low mesenchymal-like cells were found to be open/active and closed/inactive, respectively. Decreased HER2 expression was correlated with increased EMT phenotype, inactivated chromatin and lower response to lapatinib. We also found that induction of EMT in the HER2-positive breast cancer cell line BT474 resulted in downregulated HER2 expression and reduced trastuzumab binding. Our results suggest that ERBB2 gene silencing by epigenetic regulation during EMT may be a mechanism of de novo resistance of HER2-positive breast cancer cells to trastuzumab and lapatinib.

## 1. Introduction

The HER2 receptor tyrosine kinase (encoded by the ERBB2 gene) is a 1255 amino acid transmembrane epidermal growth factor receptor with tyrosine kinase activity and a molecular weight of 185 kDa [1,2]. HER2 contains four extracellular domains (ECD; domains I-IV; amino acids 1–641), an extracellular juxtamembrane region (EJM; amino acids 642–652) a transmembrane domain (TM; amino acids 653–675), an intracellular juxtamembrane region (IJM; amino acids 676–730), an intracellular tyrosine kinase domain (TK; amino acids 731–906), and a C-terminal tail (amino acids 907–1255) [3,4,5,6]. The ECD of HER2 consists of four subdomains including two leucine-rich domains (domains I and III) and two cysteine-rich domains (domains II and IV) containing receptor dimerization motif [3,4,5,6]. HER2 does not require a ligand for activation and is able to dimerize with another HER2 protein (homo-dimerization) or other HER family receptors (HER1/EGFR and HER3; hetero-dimerization). HER2 dimerization causes activation of its kinase domain which phosphorylates downstream mediators and promotes the receptor tyrosine kinase signaling cascades (PI3K/Akt, PLC-ɣ and MAPK pathways).

Overexpression of HER2 occurs in 20–30% of all breast cancer tumors (known as HER2-positive breast cancers), mostly due to ERBB2 gene amplification [2,7,8,9]. HER2 overexpression results in over-activation of downstream PI3K/Akt, PLC-ɣ and MAPK pathways leading to increased tumor cell growth, survival, motility, and invasion [10]. Targeting HER2 by small molecule inhibitors and monoclonal antibodies in patients with HER2-positive breast cancer results in significant tumor regression [11,12]. Lapatinib is a small molecule dual inhibitor of tyrosine kinase activity of HER2 and EGFR, and trastuzumab (Herceptin) and pertuzumab (Perjeta) are anti-HER2 humanized monoclonal antibodies targeting the ECD of HER2 [11,13,14]. Lapatinib, trastuzumab, and pertuzumab are all approved by the FDA to treat patients with early-stage and metastatic HER2-positive breast cancer as an adjuvant in combination with taxane therapy [11,13,14].

Acquired resistance to HER2-targeted therapies is a big obstacle in the treatment of HER2-positive breast cancer. Approximately 60–70% of HER2-positive breast cancer patients develop de novo resistance to trastuzumab; this is partially due to the loss of HER2 expression in their tumor cells during treatment [11,12,15]. A suggested mechanism of trastuzumab resistance is cleavage and shedding of HER2. Cleavage at the EJM region will result in shedding of the extracellular domains of HER2 while p95HER2 remains in the plasma membrane. In the case of cleavage at the IJM region, the intracellular domains of HER2 will shed and the extracellular domains will remain at the plasma membrane [11,12,16,17]. If either cleavage event occurs during epithelial-mesenchymal transition (EMT) in HER2-positive breast cancer, tumor cells resistant to trastuzumab but still sensitive to lapatinib will emerge [12]. Notably, little is known about the exact mechanism of lapatinib resistance in HER2-positive breast cancer.

We previously reviewed our hypothesis that suggests a role for EMT in development of de novo resistance to HER2-targeted therapies [12]. Generally, epithelial-like cells highly express HER2, whereas mesenchymal cells are HER2-negative or HER2-low. Mesenchymal-like cells show resistance to trastuzumab, suggesting that trastuzumab resistance may be linked to EMT. Here we hypothesize that wide-scale epigenetic reprogramming during EMT could be the mechanism of ERBB2 gene silencing and development of resistance to HER2-targeted agents. The aim of this study is to understand the chromatin-based epigenetic mechanism of ERBB2 gene silencing during EMT and its association with anti-HER2 therapy resistance.

### 1.1. Materials and Methods

#### 1.1.1. Antibodies

Trastuzumab (Herceptin^®^) was purchased from Roche (Basel, Switzerland). Mouse monoclonal anti-human Vimentin (V9; cat# sc-6260) and FITC-conjugated rabbit anti-mouse (cat# sc-358916) antibodies were purchased from Santa Cruz Biotechnology (Dallas, TX, USA). Rhodamine (TRITC)-conjugated donkey anti-human IgG (Cat# 709-025-149) was purchased from Jackson ImmunoResearch (West Grove, PA, USA).

#### 1.1.2. Cell Culture

The HER2-positive breast cancer cell line BT474 was purchased from American Type Culture Collection (ATCC; Manassas, VA, USA). The cells were cultured in Dulbecco’s modified Eagle’s medium (DMEM) medium supplemented with 10% fetal bovine serum (FBS) and antibiotics including penicillin (100 U/mL) and streptomycin (100 μg/mL) and were maintained at 5% CO_2_ atmosphere at 37 °C. For treatment experiments, appropriate number of cells were seeded in DMEM medium containing 10% FBS and were cultured for 24 h. The cells then were starved overnight (16 h) at DMEM containing 1% FBS before treatment and then were cultured in starvation medium containing the test agent.

#### 1.1.3. EMT Induction

A number of 2 × 10^4^ BT474 cells were seeded in 24-well plates and cultured in DMEM medium containing 10% FBS and 1× StemXVivo EMT-inducing supplement containing recombinant human Wnt-5a, recombinant human TGF-β1, anti-human E-cadherin, anti-human sFRP-1 and anti-human Dkk-1 antibodies (cat# CCM017; R&D Systems; Minneapolis, MN, USA) for 15 days. After treatment, successful EMT induction was observed by monitoring cell morphology and immunofluorescence staining of Vimentin.

#### 1.1.4. Immunofluorescence Staining Assay

The indirect double-immunofluorescence staining was done as described previously [18]. BT474 cells were cultured on 25 mm coverslip glasses in 24-well plate for 24 h and then were treated with 10 μg/mL trastuzumab for an hour. Afterward, the coverslips were washed with ice-cold PBS and the cells were fixed by incubation in −20 °C methanol for 5 min. The coverslips were then washed with TBS and blocked in coverslip blocking buffer (1% BSA solution in TBS) for 1 h. After blocking, the coverslips were incubated in 2 µg/mL primary antibody (anti-human Vimentin) solution for 1 h. The coverslips were washed and then incubated in 1 µg/mL FITC-conjugated and/or 1 µg/mL TRITC-conjugated secondary antibodies solutions for 1 h in dark. Afterward, the coverslips were washed with TBS and were then incubated in 1 µg/mL DAPI solution for 5 min. The coverslips were mounted on microscope slides, sealed by nail polish, and observed under a fluorescence microscope using FITC and TRITC channels.

#### 1.1.5. Cancer Genomic Data

RNA-seq and expression Z-scores of 1904 breast cancer tumors studied by METABRIC study were obtained from and analyzed using cBioPortal cancer genomics database [19,20] available at http://cbioportal.org/index.do (accessed on 14 August 2021).

All mRNA expression and methylation data from cell lines were obtained from GEO database available at https://www.ncbi.nlm.nih.gov/geo (accessed on 14 August 2021). GEO series and samples accession IDs of analyzed data are shown in Appendix A.

ChIP-seq data were obtained from GEO and Cistrome Data Browser [21] available at http://cistrome.org/db/# (accessed on 14 August 2021) and ChIP-seq data were visualized by using WashU Epigenome Browser [22] available at “https://epigenomegateway.wustl.edu (accessed on 14 August 2021)”. Cistrome DB and GEO samples accession IDs of analyzed ChIP-seq data are shown in Appendix A.

IM-PET promoter-enhancer interaction data were obtained from 4Dgenome database [23] available at https://4dgenome.research.chop.edu (accessed on 14 August 2021).

#### 1.1.6. Statistical Analysis and Data Visualization

GEO array expression data were analyzed by Affymetrix Transcriptome Analysis Console (TAC) 3.0 software (Affymetrix Inc., Santa Clara, CA, USA). Circus plots were created by Circa software, respectively. All figure layouts were prepared using Adobe Photoshop CS6 (San Jose, CA, USA). Data were statistically analyzed by two-tailed student’s t-test and analysis of variance (ANOVA) using Prism v.6 software (GraphPad Software, La Jolla, CA, USA). Data were presented as mean and SD. *p* < 0.050 was considered as statistically significant.

## 2. Results

### 2.1. Low ERBB2 Gene Expression in Mesenchymal-like Breast Tumors

To investigate whether the expression level of the ERBB2 gene is correlated with the expression of EMT marker genes, we analyzed the RNA-seq expression of the ERBB2 gene, 12 epithelial marker genes (ALCAM, CD24, CDH1, F11R, FOXA1, KRT7, KRT8, KRT18, KRT19, MUC, NECTIN2, NECTIN4) as well as 12 mesenchymal marker genes (CD44, CTNNB1, FOXC1, MYC, NOTCH1, NOTCH2, SNAI2, SOX10, TWIST2, VIM, ZEB1, ZEB2) in 1904 breast cancer tumor samples included in a METABRIC study [24]. We used cBioPortal portal [20] to investigate correlations between the mRNA levels of ERBB2 and the EMT markers in each tumor sample. These results showed that the expression of the ERBB2 gene was positively and negatively correlated with the expression of the epithelial phenotype (Figure 1A) and the mesenchymal phenotype (Figure 1B) marker genes, respectively. This suggests that the expression of the ERBB2 gene in epithelial-like breast cancer cells is higher than that of mesenchymal-like breast cancer cells. In other words, mesenchymal breast cancer cells exhibit low ERBB2 gene expression compared to epithelial-like breast cancer cells.

### 2.2. Similar CpG Methylation Signature of ERBB2 Promoter in Epithelial-like and Mesenchymal-like Breast Cancer Cells

To study the mechanism behind differential ERBB2 gene expression for epithelial and mesenchymal breast cancer cells, we investigated promoter CpG island methylation signatures of the ERBB2 gene in breast cancer cell lines with high ERBB2 expression (BT474, HCC-1954, MDA-MB-453, SKBR3), and those with low ERBB2 expression (BT20, MCF7, MDA-MB-231, MDA-MB-468, SUM-159PT, T47D). Array expression and array methylation data of the cells were obtained from the GEO database (Accession number: GSE44838 [25]). The result showed a positive correlation between the expression of ERBB2 and FOXA1 (epithelial-like cell marker), and a negative correlation between ERBB2 expression and the expression of FOXC1 (mesenchymal-like cell marker) in all cell lines except HCC-1954 (Figure 1C). Despite the varied ERBB2 expression levels between the cell lines, no significant difference was found between the cell lines in terms of CpG island methylation (Figure 1D). These results show that low ERBB2 expression levels in the mesenchymal-like cells are not due to promoter CpG island methylation.

### 2.3. Different ERBB2 Chromatin Signature in Epithelial-like and Mesenchymal-like Breast Cancer Cells

Since DNA-level epigenetic regulation (promoter CpG island methylation) has no role in the expression of ERBB2, we hypothesized that chromatin remodeling may control ERBB2 expression in the epithelial-like and mesenchymal-like breast cancer cells. To test this, we first studied enrichment of transcription factors at ERBB2 chromatin in the cells identified by ChIP-seq (indexed in ChIPbase v2.0 database). The results showed enrichment of 82 transcription factors within a region 10 kbp upstream and 10 kbp downstream of the ERBB2 gene motif Y in 3740 human biological samples. Of 82 transcription factors, eight (CDX2, FOXA1, FOXA2, KLF9, MBD3, MXI1, RUNX3, SP1) were regulators of epithelial maintenance, and 31 (ATF2, E2F1, E2F6, E2F7, EGR1, ELF2, ETS1, ETV1, FOS, FOXM1, FOXP1, FOXP2, GATA1, GATA2, GATA3, GATA6, HOXC9, JUNB, JUND, KDM5A, MAX, MAZ, MYC, MZF1, NANOG, RELA, SMAD4, STAT4, STAT5A, TEAD6, ZBTB7A) were master regulators of mesenchymal phenotype during EMT (Figure 1E,F).

We examined chromatin accessibility/activity of the ERBB2 promoter and enhancer by analyzing ATAC-seq and DNase hypersensitivity data of MCF7 (epithelial-like) and MDA-MB-231 (mesenchymal-like) cell lines. The results showed higher ATAC-seq and DNase I hypersensitivity signals at the ERBB2 promoter and enhancer chromatin of MCF7 cell lines compared to MDA-MB-231 cell line. The ERBB2 promoter and enhancer chromatin of MCF7 is more accessible/active than that of MDA-MB-231 (Figure 1G). This was correlated with lower and higher enrichment of the mesenchymal phenotype transcription factor E2F1 at the ERBB2 promoter in MCF7 and MDA-MB-231 cells, respectively (Figure 1G).

We then investigated whether different ERBB2 expression levels for epithelial-like and mesenchymal-like cells are due to differences in ERBB2 chromatin architecture. We analyzed the enrichment of open/active gene body chromatin histone marks (H2BK120ub, H3K39me3, H3K79me2), open/active promoter chromatin histone marks (H3K4me1, H3K4me3), open/active enhancer chromatin histone marks (H3K9ac, H3K27ac, H4K8ac), as well as closed/inactive promoter and enhancer chromatin histone marks (H3K9me, H3K27me3) at the ERBB2 gene chromatin in HER2-high (AU565, BT474, HCC-1954, MDA-MB-361, SKBR3) and HER2-low (MCF7, MDA-MB-231, MDA-MB-468) breast cancer cell lines. The mRNA expression levels of ERBB2, MUC1, and VIM are shown in Figure 2A. HER2-high cell lines AU565, BT474, HCC-1954, MDA-MB-361 and SKBR3 showed higher MUC1 mRNA expression. However, HER2-low cell lines MCF7 and MDA-MB-231 but not MDA-MB-468 cell line showed higher VIM expression (Figure 2A). These results support the observation that ERBB2 expression is correlated positively with the epithelial phenotype and negatively with the mesenchymal phenotype and suggest that mesenchymal-like breast cancer cells have low ERBB2 expression. ChIP-seq data of histone marks showed higher enrichment of H2BK120ub, H3K39me3 and H3K79me3 at the ERBB2 gene body in the HER2-high cell lines compared to those in the HER2-low cell lines (Figure 2B). The enrichment of open/active promoter chromatin marks H3K4me2 and H3K4me3 at the ERBB2 promoter chromatin in EHR2-high cell lines were also significantly higher than those in HER2-low cell lines (Figure 2C). In addition, the HER2-high cell lines showed relatively higher enrichment levels of open/active enhancer chromatin histone marks H3K9ac, H3K27ac and H4K8ac at enhancer chromatin of the ERBB2 gene when compared with the HER2-low cell lines (Figure 2D). However, the enrichment levels of closed/inactive promoter and enhancer chromatin histone marks H3K9me and H3K27me3 at ERBB2 chromatin were relatively low in HER2-high as well as in HER2-low cell lines (Figure 2E).

We then investigated the chromatin-chromatin interactions of the ERBB2 promoter in HER2-high and HER2-low breast cancer cells. For this, we studied the interactions of ERBB2 chromatin with its upstream and downstream chromatins in HCC-1954 and MCF7 breast cancer cell lines by analyzing the experimental IM-PET (Integrated Methods for Predicting Enhancer Targets) and ChIA-PET (Chromatin Interaction Analysis by Paired-End Tag Sequencing) data from the cell lines available at 4Dgenome database [23]. The ERBB2 promoter is predicted to interact with 240 target enhancer regions in the HCC-1954 cell line (Figure 2F). Of the 240 target enhancers, 134 were upstream and 106 were downstream of the ERBB2 promoter. The chromatin loop size of 106 interactions was found less than 50 kb and 18 interactions had a chromatin loop size of larger than 500 kb (Figure 2F). However, the MCF7 cell line showed the interaction of ERBB2 promoter with 11 enhancer regions which were all upstream of the ERBB2 promoter. Of 11 interactions, 10 had a chromatin loop size of less than 50 kb, and 1 interaction had a loop size of approximately 244 kb (Figure 2F).

We also analyzed the ChIP-seq H3K27ac enrichment profile of the ERBB2 interaction sites in HCC-1954 and MCF7 to examine whether the ERBB2 chromatin interactions depend on chromatin accessibility of ERBB2 or the target regions. HCC-1954 showed higher enrichment of H3K27ac at ERBB2 chromatin than MCF7. Notably, MCF7 showed higher H3K27ac enrichment at non-ERBB2 chromatin than HCC-1954. These results indicate that the number and size of promoter-enhancer chromatin loops of the ERBB2 gene in HCC-1954 are higher than those in MCF7. This was due to the higher accessibility/activity of ERBB2 chromatin in HCC-1954 compared to MCF7.

Taken together, these results suggest that chromatin activity of the ERBB2 gene governs ERBB2 gene expression in breast cancer. Epithelial-like breast cancer cells express higher levels of ERBB2 in comparison with mesenchymal-like breast cancer cells due to accessibility of ERBB2 chromatin in the epithelial-like breast cancers. ERBB2 chromatin activity is correlated with the enrichment of epithelial phenotype transcription factors and open/active chromatin histone modifications. Mesenchymal-like cells show negligible ERBB2 expression. The closed/inactive ERBB2 promoter and enhancer chromatin is correlated with both the absence of epithelial phenotype transcription factors as well as the enrichment of mesenchymal phenotype transcription factors at ERBB2 chromatin. The closed/inactive status of ERBB2 chromatin in mesenchymal-like cells is not due to inactivator histone modifications, but rather the absence of activator histone modifications at ERBB2 chromatin. Moreover, our results suggest that chromatin accessibility/activity of the ERBB2 gene is integral for interactions between ERBB2 chromatin and distance enhancer regions.

### 2.4. EMT Induces Resistance to HER2 Targeting Drugs by Downregulating HER2 Expression

We next investigated the role of EMT in the development of anti-HER2 drug resistance in HER2-positive breast cancer cells. We studied the mRNA expression of ERBB2, epithelial phenotype markers CDH1, ALCAM, FOXA1, NECTIN2, and OCLN, mesenchymal phenotype markers CDH2, FN1, FOXC1, SNAI2, and VIM, as well as matrix metalloproteinase MMP1, MMP2, MMP3, MMP9, MMP10, and MMP28 in lapatinib-sensitive and acquired lapatinib-resistant BT474 cells. The array expression profiling data was obtained from the GEO database (Series GSE16179 [26]). Lapatinib-resistant cells showed lower expression levels of ERBB2 and epithelial marker genes, and higher expression levels of the mesenchymal marker genes and MMPs when compared with the lapatinib-sensitive cells (Figure 3A). These results indicate that lapatinib-sensitive cells are epithelial-like cells with higher HER2 expression, while lapatinib-resistance cells are mesenchymal-like cells with lower HER2 expression. These results suggest that EMT induces resistance to lapatinib via downregulating HER2 expression.

To examine whether the induction of EMT reduces HER2 expression, we analyzed mRNA expression of ERBB2, epithelial phenotype markers CDH1, EPCAM, MUC1 and OCLN, mesenchymal phenotype markers CDH2, FN1, SNAI2, and VIM in the A549 cell line (HER2-high human lung cancer epithelial cell line). A549 was subjected to EMT induction by treatment with 5 ng/mL TGF-β for 0, 0.5, 1, 2, 4, 8, 16, 24 and 72 h. The array expression profiling data were obtained from the GEO database (Series GSE17708 [27]). The mRNA expression was significantly decreased for the epithelial phenotype marker genes and significantly increased for the mesenchymal phenotype marker genes, indicating induction of EMT in A549 cells. EMT induction was correlated with a significant decline in ERBB2 expression at 72 h after TGF-β treatment started (Figure 3B).

We also investigated whether EMT induction reduces trastuzumab binding. For this, we induced EMT in BT474 (HER2-high breast cancer cells) by treating the cells with an EMT-inducing supplement cocktail (containing Wnt-5a, TGF-β1, anti-human E-Cadherin antibody, anti-human sFRP1 antibody, and anti-human Dkk1 antibody) for 15 days. Induction of EMT was confirmed by monitoring cell morphology and immunofluorescence staining of Vimentin (Figure 3C). After the majority of the BT474 cells gained mesenchymal phenotype, the cells were first treated with 10 μg/mL trastuzumab for 1 h and then the trastuzumab was stained by immunofluorescence staining. Cells that underwent EMT showed lower HER2 expression as well as lower binding of trastuzumab to HER2 compared to the control cells (Figure 3C). Taken together, these results confirm that EMT of HER2-high cells can lead to resistance to HER2-targeting agents by downregulating HER2 expression particularly via epigenetic remodeling of ERBB2 chromatin.

## 3. Discussion

HER2 is an important target for the treatment of the HER2-positive subtype of breast cancers. Several HER2-targeting agents such as trastuzumab and lapatinib have been approved by the FDA to treat HER2-positive breast cancer [12]. However, the development of de novo resistance to HER2-targeting drugs has become biggest challenge for treatment. Previous studies demonstrated that the loss of expression of HER2 in tumor cells likely causes HER2-targeting drug resistance [11,12,15]. Here, we studied an epigenetic mechanism of HER2 downregulation leading to HER2-targeting drug resistance in breast cancer. We hypothesized that chromatin remodeling during EMT of HER2-high breast cancer cells could result in downregulation of ERBB2 gene expression leading to the development of drug resistance. To test this hypothesis, we investigated the expression and chromatin-based epigenetic signature of the ERBB2 gene in epithelial-like HER2-high and mesenchymal-like HER2-low breast cancer cells. We also studied the effect of EMT on HER2 expression levels and cellular response to HER2-targeting drugs. In addition, by analyzing the epigenomics data from publicly available databases we investigated the role of EMT-mediated epigenetic regulation in ERBB2 chromatin organization.

We found that the expression of EMT markers and inducers is negatively correlated with HER2 expression, and positively correlated with trastuzumab and lapatinib resistance. HER2 expression in epithelial-like breast cancer cells was significantly higher than that in mesenchymal-like breast cancer cells. This is due to open/active chromatin at the ERBB2 gene in epithelial-like cells, as well as closed/inactive chromatin at the ERBB2 gene in mesenchymal-like breast cancer cells. This was also correlated with lower and higher enrichment levels of EMT regulator transcription factors at the *cis*-regulatory regions of the ERBB2 gene in epithelial-like and mesenchymal-like breast cancer cell lines respectively. Our analysis also showed downregulated HER2 expression and upregulated EMT marker genes in BT474 cells resistance to lapatinib compared to lapatinib-sensitive cells. Induction of EMT in HER2-high epithelial-like breast cancer cells resulted in the downregulation of HER2 and decreased rate of trastuzumab binding to the cells. Our results suggest that EMT of HER2-positive breast cancer cells results in abrogation of HER2 expression by chromatin-based epigenetic silencing of the ERBB2 gene that leads to trastuzumab resistance.

CD44+/CD24− cells are identified as mesenchymal-like BCSCs that localize at the tumor margins and are to blame for metastasis, whereas ALDH+ cells are defined as differentiated epithelial-like BCSCs that are found in deeper sites of tumors and have more proliferative properties [12,28]. CD44+/CD24− cells isolated from non-tumorigenic human mammary epithelial cells that have undergone induced EMT exhibited many BCSCs properties including the ability to form mammospheres [29]. CD44+/CD24− BCSCs isolated from breast tumors, however, expressed a low level of epithelial phenotype but high levels of EMT markers [29]. Several clinical studies have demonstrated upregulated EMT in HER2-positive metastatic breast cancers [30,31]. Activation of HER2-mediated signaling in epithelial breast cells has been shown to increase the expression of mesenchymal phenotype genes Vimentin, N-cadherin, and Integrin-α5 and decrease the expression of epithelial phenotype genes E-cadherin and Desmoplakin. However, other reports suggest that downregulation of E-cadherin is not necessary for HER2-mediated induction of EMT [32,33]. In addition, overactivation of HER2 in epithelial breast cancer cells has been shown to induce EMT and maintain mesenchymal phenotype [34].

Several studies suggest a relationship between HER2 expression, upregulation of mesenchymal phenotypes, and development of trastuzumab resistance. Although HER2 is associated with the epithelial phenotype, sporadic reports have shown that trastuzumab-resistant breast tumors can exhibit high expression levels of HER2 as well as the mesenchymal phenotype [12]. For example, the basal breast cancer cell line JIMT-1, which is HER2-high, is highly enriched for CD44+/CD24− cells that can quickly develop de novo resistance to trastuzumab in late-passages [35,36,37]. Development of trastuzumab resistance in JIMT-1 cells coincides with increased expression of EMT genes and subpopulation of CD44+/CD24− cells [37,38,39,40,41,42,43]. JIMT-1 was shown to be composed of approximately 10% CD44+/CD24− BCSCs in initial cultures; however, this level rose to 85% at the late passages [37]. HER2 expression levels were significantly reduced in late passage cultures when compared to the early cultures, which is associated with the development of trastuzumab-resistance [37]. Thus, the resistance of HER2-high breast tumors to trastuzumab may be due to an increased population of HER2-low CD44+/CD24 mesenchymal cells at the late passages. As another example, SKBR3 cells are HER2-positive luminal/epithelial cells and are trastuzumab-sensitive; however, mesenchymal CD44+/CD24− cells derived from the same SKBR3 culture are HER2-low and highly resistant to trastuzumab and lapatinib [44]. In addition, EMT can result in the development of resistance of HER2-positive tumors to trastuzumab-mediated ADCC by increasing the subpopulation of HER2-low mesenchymal cells [45].

It has been demonstrated that HER2 itself induces EMT of breast cancer cells via upregulating stemness pathways. This suggests a negative feedback loop between HER2 and EMT that gives us important clues about the trastuzumab-responsive HER2-positive breast cancer that develops resistance to trastuzumab. Overall, our results demonstrate that EMT of HER2-positive breast cancer cells causes the emergence of a subpopulation of mesenchymal-like cells which are HER2-low/negative and highly resistant to trastuzumab and lapatinib. The significant depletion of HER2 in the mesenchymal-like cells inside the tumor can take place by chromatin-based epigenetic silencing of the ERBB2 gene during EMT. These results suggest that ERBB2 gene silencing by epigenetic regulation during EMT is an authentic mechanism of downregulated HER2 in the mesenchymal-like cells and the main mechanism of resistance of HER2-positive breast cancer cells to trastuzumab and lapatinib. Here, we suggest that EMT may be the major mechanism of resistance to HER2-targeting therapeutics trastuzumab and lapatinib. Therefore, EMT and mesenchymal-like cells, particularly cancer stem cells, can serve as bona fide targets to overcome drug resistance in breast cancers. As a future direction, we recommend investigating the mechanism of the negative feedback loop to understand how HER2 overexpression induces EMT, and how EMT causes ERBB2 gene silencing, leading to the emergence of tumor cells resistant to HER2-targeted therapies.

## Figures and Tables

**Figure 1 life-11-00868-f001:**
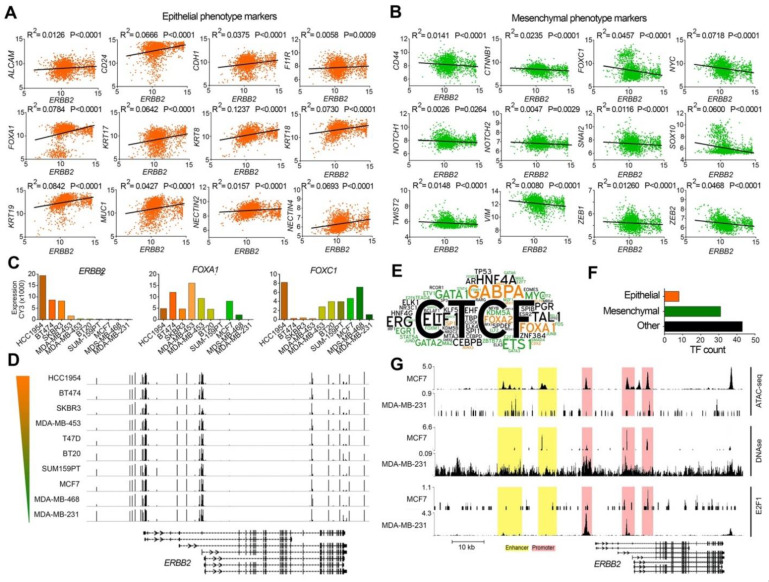
Different ERBB2 expression and chromatin signature in epithelial-like and mesenchymal-like breast cancer cells. (**A**,**B**) Correlation between the mRNA expression of ERBB2 and epithelial (**A**) and mesenchymal (**B**) phenotype marker genes in human breast cancer tumors. The normalized RNA-seq expression data from 1904 breast tumors studied by METABRIC study [24] are presented by Z-score fold changes RNA-seq expression (v2 RSEM). (**C**) mRNA expression levels of ERBB2, FOXA1, and FOXC1 genes in 10 breast cancer cell lines. (**D**) Methylation levels of promoter CpG islands in the cell lines. The color gradient bar indicates HER2 expression level. Genomic coordinate: chr17:37,834,978–37,897,500 (GRCh37/hg19 assembly). (**E**) Word cloud diagram of transcription factors (TFs) detected bound to the ERBB2 gene at 10 kb upstream and downstream of motif Y. The different number of binding sites for each TF at the query region is illustrated as a different word size. TFs promoting epithelial and mesenchymal phenotypes are illustrated in orange and green colors respectively. (**F**) The number of the identified TFs based on their function in EMT. (**G**) ChIP-seq enrichment value peaks of ATAC-seq, DNase I hypersensitivity and E2F1 at ERBB2 gene regulatory regions in MCF7 and MDA-MB-231 cell lines. Genomic coordinate: chr17:39,643,771–39,735,523 (GRCh38/hg38 assembly).

**Figure 2 life-11-00868-f002:**
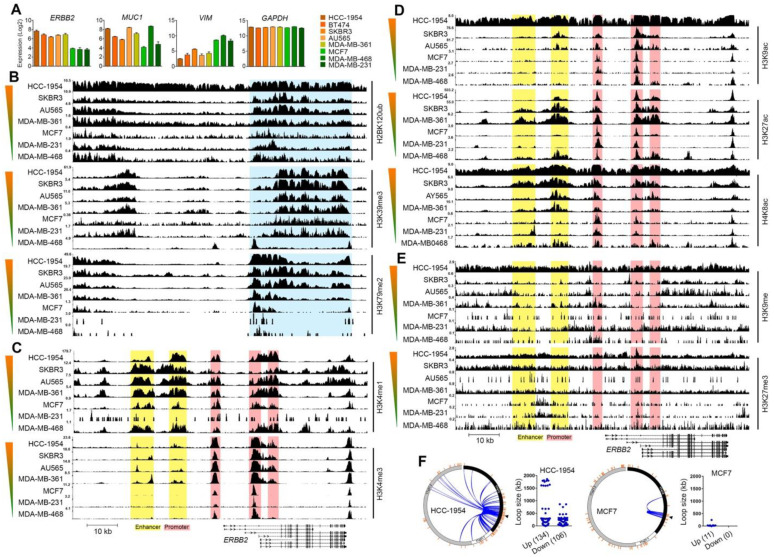
Histone marks of ERBB2 chromatin in epithelial-like and mesenchymal-like breast cancer cells. (**A**) mRNA expression levels of *ERBB2,* epithelial marker gene MUC1, mesenchymal marker VIM gene and GAPDH in HER2-high (HCC-1954, BT474, SKBR3, AU464, MDA-MB-361) and HER2-low (MCF7, MDA-MB-231, MDA-MB-468) breast cancer cell lines. (**B**–**E**) ChIP-seq enrichment of open/active gene body (**B**), open/active promoter (**C**), open/active enhancer (**D**), closed/inactive promoter and enhancer (**E**) chromatin histone marks of the ERBB2 gene and upstream region in the cell lines. Color gradient bars indicate the HER2 expression level. Genomic coordinate: chr17:39643771-39735523 (GRCh38/hg38 assembly). (**F**) Circle interaction and scatter plots of IM-PET promoter-enhancer interactions of ERBB2 chromatin in HCC-1954 (HER2-high) and MCF7 (HER2-low) cell lines. Scatter plots illustrate chromatin loop size of ERBB2 promoter-enhancer interactions. Arrowheads show position of ERBB2 TSS.

**Figure 3 life-11-00868-f003:**
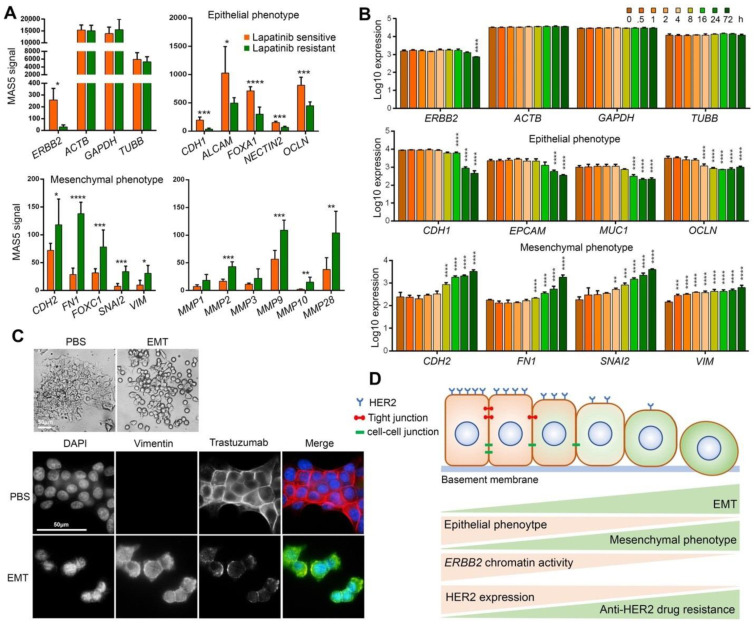
EMT induces trastuzumab resistance by downregulating HER2 expression. (**A**) mRNA expression levels of ERBB2 and housekeeping genes, epithelial phenotype marker genes, mesenchymal phenotype marker genes and matrix metalloproteinases in lapatinib sensitive and resistant BT474 cells. (**B**) mRNA expression levels of ERBB2 and housekeeping genes, epithelial phenotype marker genes and mesenchymal phenotype marker genes in TGF-β-mediated EMT-induced A549 cells. * *p* < 0.05; ** *p* < 0.01; *** *p* < 0.001; **** *p* < 0.0001. (**C**) Epithelial morphology of PBS-treated BT474 cells and mesenchymal morphology of EMT-induced BT474 cells (top). Immunofluorescence staining of Vimentin and trastuzumab in EMT-induced BT474 cells treated with 10 μg/mL trastuzumab for 1 h (Bottom). (**D**) Schematic summary of findings. EMT of HER2-positive breast cancer cells increases trastuzumab resistance by chromatin-based epigenetic downregulation of HER2 expression. Increased EMT and mesenchymal phenotype is correlated with decreased expression of epithelial phenotype (including tight junctions and cell-cell junction proteins) and increased mesenchymal phenotype, decreased chromatin accessibility/activity including reduced enrichment of open/active chromatin marks (H3K4me and H3Kac as examples), increased enrichment of closed/inactive chromatin marks (H3K9me as an example), decreased HER2 expression and eventually increased resistance to anti-HER2 drugs including trastuzumab.

## Data Availability

Data available in a publicly accessible repository. The data presented in this study are openly available by accession numbers provided in Appendix A.

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
