# Peer review of "Epigenetic Silencing of HER2 Expression during Epithelial-Mesenchymal Transition Leads to Trastuzumab Resistance in Breast Cancer"

_life, 2021, doi:10.3390/life11090868_

Round 1
Reviewer 1 Report
This work explores mechanisms that may be responsible for the loss of HER2 expression during treatment with HER2-targeting therapies in HER2-positive breast cancers (accounting for 25% of breast cancer cases), and that has been linked to the development of resistance. By analyzing publicly available genomic databases, the authors investigate if the epithelial-to-mesenchymal transition (EMT) abrogates HER2 expression by epigenetic silencing of ERBB2 gene, a mechanism that may lead cells to acquire resistance to HER2-targeting therapies. They found that HER2 expression is positively and negatively correlated with the expression of epithelial and mesenchymal markers, respectively. The ERBB2 chromatin was found open in HER2-high epithelial-like breast cancer cells and close in HER2-low mesenchymal-like cells. In addition, inactivated chromatin was found correlated with resistance to lapatinib. They also show that the induction of EMT in BT474 (HER2-positive) causes downregulation of HER2 expression and reduces trastuzumab binding. Thus, results suggest that ERBB2 silencing during EMT may be a mechanism of causing resistance of HER2-positive breast cancer cells to trastuzumab and lapatinib. The results are relevant for publication, even having some limitations such as the functional studies being performed in one single cell line and determining just the resistance to trastuzumab. The involvement of epigenetic regulation is also not fully explored. Nevertheless, the manuscript can be improved especially in terms of English, and be considered suitable for publication with no further experiments. There are numerous corrections to make throughout the manuscript and will next give some examples
- Name of the authors should be followed by numbers superscript
- Mistakes exist in the affiliations
- Several mistakes were found in the abstract and this part can be further improved to fully deliver the main results and their importance.
- “de novo” and “cis” should be written in italic, “HER2 positive” should be “HER2-positive” throughout the manuscript.
- Some articles are required throughout the manuscript (“the”, “a”, etc). I advise the use of “Grammarly”, an online tool that helps to identify easily this type of mistake, as well as the lack of commas that also need to be corrected throughout the text.
INTRODUCTION
- 4) In the introduction, the authors need to add
- Some sentences have problems:
- “Targeting HER2 by small molecule inhibitors and monoclonal antibodies is the current therapy for HER2-positive breast cancer that outcome significant tumor regression in the patients [11,12].”
- To big: “Lapatinib, a small molecule dual inhibitor of tyrosine kinase activity of HER2 and EGFR, and trastuzumab (Herceptin) and pertuzumab (Perjeta) which are anti-HER2 humanized monoclonal antibodies targeting ECD of HER2 approved by FDA to treat patients with early-stage and metastatic HER2-positive breast cancer as an adjuvant in combination with taxane therapy [11,13,14].”
- “Both cleavage cases that can take place by proteinases during epithelial-mesenchymal transition (EMT) of HER2-positive breast cancer, that leads to emergence of tumor cells resistant to trastuzumab but still sensitive to lapatinib [12].”
- “Here hypothesize that wide-scale epigenetic reprogramming during EMT could be the mechanism of ERBB2 gene silencing and development of resistance to HER2-targeted agents.”
METHODS
- There are mistakes in verbs: “was” instead of “were” and vice-versa.
- " Word could diagrams” is not often used as a designation
FIGURE
- in figure 1 legend there is an extra “in”
RESULTS
- There are problems with some sentences
- “Result showed enrichment 191 of a totally 82 transcription factors at the region of 10 kbp upstream and 10 kbp down- 192 stream of ERBB2 gene motif Y in 3,740 human biological samples.”
- “Of 240 target enhancers, 134 were at upstream and 106 were at downstream of ERBB2 promoter. ….However, MCF7 cell line showed the interaction of ERBB2 promoter with 11 enhancer region which all were at upstream of ERBB2 promoter. …”
- “Whereas MCF7 cells showed higher H3K27ac enrichment at non-ERBB2 chromatin than 263 that in HCC-1954 cell lines.”
- “The array expression profiling data obtained from GEO 288 database (Series GSE16179 [26]).”
- “The array expression 316 profiling data obtained from GEO database (Series GSE17708 [27]).”
- “Results showed lower HER2 expression as well as lower binding of trastuzumab to HER2 in the cells underwent EMT compared to the control cells 330 (Figure 3C).”
- “lapatinib sensitive cells” should be “lapatinib-sensitive cells”, “EMT inducing media” should be “EMT-inducing media”
DISCUSSION
- The authors can improve the beginning of the discussion
- References are lacking
- “Previous studies demonstrated the loss of expression of HER2 on tumor 339 cells as a major reason of HER2-targeting drug resistance.”
Author Response
Thank you for your careful reviewing. The manuscript has been revised carefully by Dr. Corbin Black a native English expert in the field. All mistakes and grammatical errors were corrected as required. Some sentences were reworded to better understand. References were added to the sentence.
Reviewer 2 Report
In the first part, the authors were interested to find an explanation for primary resistance to HER2 targeted therapy and hypothesize that epigenetic reprogramming during EMT explains the low levels of HER2 in mesenchymal-like cells. This is a very interesting research question and if it holds has important therapeutic implications. The story is written in a logical order and is mostly easy to follow. They use many different and elegant approaches to address their hypothesis. The authors use many already available data sets. This is interesting and shows the potential of already available data! However, there is one major limitation, which concerns me: The authors used in their study cell lines with HER2 amplification. Since these are mostly epithelial, epigenetic changes are probably not the main cause of the different HER2 expression levels between epithelial and mesenchymal cell lines. Therefore, I partially put the conclusion, the authors draw, in question. Do they have a counter-argument to that? Of note: therefore, I rated the scientific soundness as low.
In a second part, the authors focused on acquired resistance to HER2-targeted therapy. Here, the authors convincingly demonstrate that lapatinib-resistant BT474 cells have a more mesenchymal phenotype than their sensitive counterparts. The other way around, EMT induction in BT474 cells induces resistance to trastuzumab. However, whether it is due to epigenetic changes, remains to be elucidated.
Minor concerns/questions:
- Figure 1 A:
- too small
- Correlation/Statistics is missing => what is r? what is p?
- too small
- Provide references for epithelial or mesenchymal master regulators (line 193 to 198)
- Line 209 to 233: This paragraph is not easy to follow. Consider re-writing.
- Figure 1 G
- Reference is missing to E2F1 is as a “mesenchymal phenotype transcription factor
- Figure 3 C
- Are these cells also resistant to lapatinib? In the introduction, the authors mention that cleavage by metalloproteases could lead to trastuzumab resistance, whereas cells are still sensitive to lapatinib.
Major concerns/questions (mostly concerns the use of HER2 amplified cell lines):
Figure 1 C
- Some cell lines have a HER2 amplification (SKBR3, HCC1954, BT474), which probably explains the higher HER2 mRNA levels of these cell lines. If the authors would like to make a point on epigenetic regulation of HER2 levels, it is questionable to use HER2 amplified cell lines. Do the authors have any counter-argument? At least the cell line status (ER/PR/HER2 status) should be given in the text that readers can judge the results appropriately.
- Why only one marker for epithelial or mesenchymal phenotype is chosen to correlate with HER2 mRNA levels in the cell lines? Why exactly are these markers? Are these markers sufficiently determining an epithelial or mesenchymal phenotype respectively?
- Figure 1 G
- A very elegant experiment, however, why did the authors choose MCF7 as an epithelial cell line to analyze for chromatin accessibility at the HER2 promoter? They express low levels of HER2 mRNA based on Figure 1 C. Do they express significantly more HER2 than MDA-MB-231?
- Figure 2
- See remark to Figure 1C. If the authors would like to make a point on epigenetic regulation of HER2 mRNA levels, it is questionable to use HER2 amplified cell lines to do so.
- Line 268 to 280: I do not fully support these conclusions. Indeed, chromatin accessibility/activity at the HER2 locus is higher in more epithelial-like cells. This has nicely been demonstrated by the authors with several different approaches. However, is it the main reason for higher HER2 levels? Many of the epithelial-like cells, used in the study, harbor a HER2 amplification (SKBR3, HCC-1954, AU565, BT474).
Author Response
In the first part, the authors were interested to find an explanation for primary resistance to HER2 targeted therapy and hypothesize that epigenetic reprogramming during EMT explains the low levels of HER2 in mesenchymal-like cells. This is a very interesting research question and if it holds has important therapeutic implications. The story is written in a logical order and is mostly easy to follow. They use many different and elegant approaches to address their hypothesis. The authors use many already available data sets. This is interesting and shows the potential of already available data! However, there is one major limitation, which concerns me: The authors used in their study cell lines with HER2 amplification. Since these are mostly epithelial, epigenetic changes are probably not the main cause of the different HER2 expression levels between epithelial and mesenchymal cell lines. Therefore, I partially put the conclusion, the authors draw, in question. Do they have a counter-argument to that? Of note: therefore, I rated the scientific soundness as low.
In a second part, the authors focused on acquired resistance to HER2-targeted therapy. Here, the authors convincingly demonstrate that lapatinib-resistant BT474 cells have a more mesenchymal phenotype than their sensitive counterparts. The other way around, EMT induction in BT474 cells induces resistance to trastuzumab. However, whether it is due to epigenetic changes, remains to be elucidated.
Minor concerns/questions:
- Figure 1 A:
- too small
- Correlation/Statistics is missing => what is r? what is p?
Authors' response: We tried to illustrate the figures and words as readable as much as possible. We had limited space for figures in the layout due to many panels. However, since the paper will be published online, we do not think that readers will have difficulty in reading the figures. R-squared and P values were added to Figure 1A.
- Provide references for epithelial or mesenchymal master regulators (line 193 to 198)
Authors' response: The information was collected by reviewing the literature. Currently, no database for the classification of transcription factors in terms of regulation of epithelial and mesenchymal phenotype. The transcription factors we mentioned in Figure 1E are well-studied proteins. Due to limited space for the number of references, we were unable to acknowledge all the reference studies. There are plentiful reference papers available in PubMed and Google Scholar if you search.
- Line 209 to 233: This paragraph is not easy to follow. Consider re-writing.
Authors' response: The sentence was re-written.
- Figure 1 G
- Reference is missing to E2F1 is as a “mesenchymal phenotype transcription factor
Authors' response: E2F family of transcription is known as regulators of stemness maintenance and inducer of EMT. There are some references:
- Chong et al. E2f1-3 switch from activators in progenitor cells to repressors in differentiating cells. Nature. 2009 Dec 17;462(7275):930-4.
- Xu et al. E2F1 Suppresses Oxidative Metabolism and Endothelial Differentiation of Bone Marrow Progenitor Cells. Circ Res. 2018 Mar 2;122(5):701-711.
- Wang et al. Transcription factor E2F1 promotes EMT by regulating ZEB2 in small cell lung cancer. BMC Cancer. 2017 Nov 7;17(1):719.
- Julian and Blais. Transcriptional control of stem cell fate by E2Fs and pocket proteins. Review Front Genet. 2015 Apr 28;6:161.
- Figure 3 C
- Are these cells also resistant to lapatinib? In the introduction, the authors mention that cleavage by metalloproteases could lead to trastuzumab resistance, whereas cells are still sensitive to lapatinib.
Authors' response: HER2 cleavage is not a common mechanism of trastuzumab resistance. Some reports showed that HER2 cleavage by MMPs results in trastuzumab resistance while lapatinib sensitivity can preserve. In Figure 3C, we did not test how the cells are sensitive to lapatinib. We even did not test their sensitivity to trastuzumab. We only showed that binding trastuzumab to HER2-positive cells dramatically decreases after EMT that is due to downregulation of the HER2 ectodomain.
Major concerns/questions (mostly concerns the use of HER2 amplified cell lines):
Figure 1 C
- Some cell lines have a HER2 amplification (SKBR3, HCC1954, BT474), which probably explains the higher HER2 mRNA levels of these cell lines. If the authors would like to make a point on epigenetic regulation of HER2 levels, it is questionable to use HER2 amplified cell lines. Do the authors have any counter-argument? At least the cell line status (ER/PR/HER2 status) should be given in the text that readers can judge the results appropriately.
Authors' response: Yes. It is well-known that overexpression of HER2 in cell lines and tumors is mostly due to ERBB2 gene amplification. This fact, however, does not explain the significant correlation of epithelial and mesenchymal phenotype with HER2 expression (Figure 1A). Please remember that breast cancer tumors that are clinically classified as HER2-positive subtype (based on HER2 gene amplification and histochemistry staining) lose their HER2 expression months after trastuzumab treatment starts and turn to HER2-low and trastuzumab-resistant. This is the main question of our study. Here, we are not pursuing a reason for high or low HER2 expression, but we are studying a potential mechanism of HER2 downregulation in HER2-positive cells and tumors regardless of HER2 gene amplification status or any other genetic background. Figure 3C is a good example. BT474 cells that have ERBB2 gene amplification lose HER2 overexpression after EMT.
Why only one marker for epithelial or mesenchymal phenotype is chosen to correlate with HER2 mRNA levels in the cell lines? Why exactly are these markers? Are these markers sufficiently determining an epithelial or mesenchymal phenotype respectively?
Authors' response: Of course, there are other markers for epithelial and mesenchymal phenotypes that mostly are structural proteins (such as E-cadherin for epithelial and Vimentin for mesenchymal phenotypes). The main reason that we show the expression of FOXA1 and FOXC1 as markers is that these proteins are master transcriptions in maintaining epithelial (FOXA1) and mesenchymal (FOXC1) phenotypes in breast cancer. However other markers support the correlation as well. The accession numbers of the original datasets are given in Supplementary Tables 1 and 2. Simple expression analysis by the reader will reveal a correlation between the expression of ERBB2 and that of other marker genes.
- Figure 1 G
- A very elegant experiment, however, why did the authors choose MCF7 as an epithelial cell line to analyze for chromatin accessibility at the HER2 promoter? They express low levels of HER2 mRNA based on Figure 1 C. Do they express significantly more HER2 than MDA-MB-231?
Authors' response: Choosing MCF7 was because of its higher HER2 expression compared to MDA-MB-231. We would like to look HER2-high cell lines as well, but unfortunately, among the cell lines with a HER2 expression higher than that of MDA-MB-231, only data of the MCF7 cell line is available in public databases.
- Figure 2
- See remark to Figure 1C. If the authors would like to make a point on epigenetic regulation of HER2 mRNA levels, it is questionable to use HER2 amplified cell lines to do so.
- Line 268 to 280: I do not fully support these conclusions. Indeed, chromatin accessibility/activity at the HER2 locus is higher in more epithelial-like cells. This has nicely been demonstrated by the authors with several different approaches. However, is it the main reason for higher HER2 levels? Many of the epithelial-like cells, used in the study, harbor a HER2 amplification (SKBR3, HCC-1954, AU565, BT474).
Authors' response: We think that ERBB2 gene amplification and copy number variation do not have a big effect on the enrichment of histone modification marks at ERBB2 promoter and enhancer. Please pay attention to the scales of ChIP-seq peaks. For example, the scale of H3Kme1 peaks in HCC-1954 is 179.7, while this value for MDA-MB-231 is 1.7. In addition, the Difference between HER2-high and HER2-low cell lines is considered significant in terms of peaks at ERBB2 enhancer (yellow highlight). If you see the peaks of open enhancer histone marks (H3Kme3, H3K9ac, and H3K27ac) MCF7 and MDA-MB-231 cells had almost zero enrichment at the enhancers. MDA-MB-468 cells also did not show H3Kme3, H3K9ac enrichment at ERBB2 enhancers. Regardless of the ERBB2 gene copy number, the enhancer should show at least H3Kme3, H3K9ac, and H3K27ac peaks if their ERBB2 enhancer chromatin were open. In addition, as mentioned in the manuscript, we confirmed the ChIP-seq data by investigating the chromatin interactions of the ERBB2 gene in HCC-1954 and MCF7 cells (Figure 2F). Chromatin interactions are a good indicator of chromatin activity and a key controller of gene transcription. As shown in Figure 2F, the interaction of ERBB2 chromatin with upstream and downstream chromatin in HCC-1954 cells is significantly higher than that of MCF7 cells. This confirms that chromatin accessibly/activity controls HER2 expression in the cells.